# Cervical CT Angiography: The Advantage of Ultra-High-Resolution CT Versus Conventional HRCT

**DOI:** 10.3390/cancers16223866

**Published:** 2024-11-19

**Authors:** Junji Ito, Tsuneo Yamashiro, Hayato Tomita, Joichi Heianna, Sadayuki Murayama, Akihiro Nishie

**Affiliations:** 1Department of Radiology, Graduate School of Medical Science, University of the Ryukyus, Okinawa 903-0215, Japan; clatsune@yahoo.co.jp (T.Y.); m04149@yahoo.co.jp (H.T.); jana1150@nantoku.org (J.H.); sadayuki@med.u-ryukyu.ac.jp (S.M.); nishie_a@med.u-ryukyu.ac.jp (A.N.); 2Diagnostic Radiology, Yokohama City University Hospital, Kanagawa 236-0004, Japan; 3Department of Radiology, St. Marianna University School of Medicine, Kanagawa 216-8511, Japan

**Keywords:** computed tomography, CT angiography, ultra-high-resolution CT, cervical artery

## Abstract

Intra-arterial infusion chemotherapy is an effective therapeutic option for malignant head and neck tumors. Precise advanced information on the arteries supplying tumors enables us to achieve a good treatment effect; shorten the operation time; and minimize both radiation exposure and the use of contrast medium. Therefore, a preoperative assessment of patients is clinically important. Although ultra-HRCT (U-HRCT) provides improved CT images with 0.25-millimeter slice thickness reconstruction, the usefulness of U-HRCT for CT angiography (CTA) has also been recently reported. We hypothesized that U-HRCT might contribute to the improvement in image quality for cervical CTA as well. Based on the investigation from the points of view of both phantom and patient studies, U-HRCT provided higher image quality in terms of visualization of cervical arteries than conventional HRCT.

## 1. Introduction

Intra-arterial infusion chemotherapy is an effective therapeutic option for malignant head and neck tumors [1,2,3,4,5,6,7,8]. In particular, a combination of intra-arterial infusion chemotherapy and external radiation therapy has been suggested as a more effective treatment for malignant head and neck tumors [1,2]. For successful intra-arterial infusion chemotherapy, it is necessary to identify all arteries supplying blood to the tumor [6]. Previously, a detailed preoperative assessment of the arteries supplying tumors using high-resolution computed tomography (HRCT) was recommended to achieve a good treatment effect, shorten the operation time, and minimize both radiation exposure and the use of contrast medium [9,10,11,12].

Recently, ultra-HRCT (U-HRCT)—which can provide improved CT images with 0.25-millimeter slice thickness reconstruction—has been clinically applied in several different imaging fields. In addition to the utility for chest and temporal bone imaging [13,14,15,16,17,18,19], the improvement in image quality using U-HRCT for CT angiography (CTA) has also been reported [20,21,22,23,24,25,26,27]. For example, Yoshioka et al. mentioned that the artery of Adamkiewicz was better depicted with U-HRCT compared to conventional HRCT (0.5-millimeter thickness) [20]. It has also been reported that fine intracranial, coronary, visceral, and peripheral arteries in the extremities are better depicted by U-HRCT than conventional HRCT [21,22,23,24,25,26,27,28,29]. Based on the advantages of U-HRCT over conventional HRCT to depict small arteries, we hypothesized that cervical CTA would also be greatly improved with U-HRCT in comparison to conventional HRCT. To the best of our knowledge, the image quality for cervical CTA using U-HRCT has not yet been investigated. 

The aim of this study was to investigate the advantages of U-HRCT over conventional HRCT for cervical CTA from the points of view of both phantom and patient studies.

## 2. Materials and Methods

This retrospective study was approved by our institutional review board, which waived the need for informed consent from patients. Opt-out opportunities were provided to all participants included in this retrospective study on the website of our department.

### 2.1. Phantom Study

#### 2.1.1. CT Scan Equipment

This phantom study was conducted using a head-neck CT phantom (No. 41309-100, Kyotokagaku, Kyoto, Japan). The phantom was made of polyurethane resin and epoxy resin and equipped with mock carotid-cerebral arteries on one side, which were shown as enhanced arteries by contrast medium on the CT scans. The phantom was scanned using a conventional 320-row HRCT scanner (Aquilion ONE, Canon Medical Systems, Otawara, Tochigi, Japan) and a 160-row U-HRCT scanner (Aquilion Precision, Canon Medical Systems), which were the same scanners used for the patient study.

#### 2.1.2. CT Scan Parameters

The CT parameters for the conventional HRCT acquisition were as follows: tube voltage = 120 kVp; tube current = 200 mA; collimation = 0.5 mm × 80 slice; rotation time = 0.5 s; imaging field of view (FOV) = 240 mm; matrix = 512 × 512; slice thickness = 0.5 mm; channel number = 896; iterative reconstruction = adaptive iterative dose reduction using three-dimensional processing (AIDR3D). For the U-HRCT acquisition, super-high-resolution mode (SHR mode) and high-resolution mode (HR mode) were scanned. The scanning and reconstruction parameters were as follows: tube voltage = 120 kVp; tube current = 200 mA; [SHR mode] collimation = 0.25 mm × 160 slice and [HR mode] collimation = 0.5 mm × 80 slice; rotation time = 0.5 s; imaging field of view (FOV) = 240 mm; matrix = 512 × 512; [SHR mode] slice thickening = 0.25 mm and [HR mode] slice thickening = 0.5 mm; channel number = 1792; iterative reconstruction = AIDR3D-enhanced (AIDR3D-e).

#### 2.1.3. Image Assessment

Since the phantom was placed at almost the same location on the bed of both CT scanners, the CT images from the two scanners were almost completely matched on the workstation (Ziostation2, Ziosoft, Tokyo, Japan). Using a density curve on the axial images, the maximum densities of the mock arteries were measured at a similar position on both conventional HRCT and two scan modes of U-HRCT, and the density curve was manually reproduced on the two CT scans. This measurement was performed at 45 points on the mock arteries: proximal, 15 points on the mock common and internal carotid artery (approximately 3.5 mm in diameter); intermediate, 15 proximal points on the middle cerebral artery (approximately 2.5–3.0 mm in diameter); peripheral, distal 15 points on the anterior and middle cerebral arteries (approximately 1.0–1.5 mm in diameter). Since the phantom did not have an external carotid artery and its branches, we measured the mock cerebral arteries for the intermediate and distal points instead. The measurement was performed by a radiologist with 4 years of experience who was unaware of the scanning method. 

Next, 3-dimensional (3D) CTA-like images were created for conventional HRCT and two scan modes of U-HRCT using the same workstation (Ziostation2). We selected 3D CTA-like images, which might be easier to understand in terms of location and anatomical structure than maximum-intensity projection images. These images were displayed with a window width of 116 and window length of 79. Five branches, in order from the proximal sites of the anterior and middle cerebral arteries, were assessed visually and recorded by two radiologists with 4 and 25 years of experience in consensus, who were unaware of the scanning method. As a result, a total of 10 branches were evaluated. Scores were recorded using a 5-point scale (score 1 = the origin of the branch was only visualized; score 2 = less than 1/3 of the branch was visualized; score 3 = 1/3 to less than 2/3 of the branch were visualized: score 4 = 2/3 or more of the branch was visualized).

### 2.2. Patient Study

#### 2.2.1. Subjects

The CTA exams were performed on the conventional HRCT scanner from January 2016 to July 2017, and subsequently, on the new U-HRCT scanner from August 2017 to May 2018. During this period, 23 consecutive patients were scanned on the conventional HRCT scanner and 21 consecutive patients were scanned on the U-HRCT scanner. Among them, those who met the following criteria were excluded: (1) One or more main branches of the external carotid artery were ligated by a previous operation (n = 2). (2) The main trunk of the external carotid artery was not clearly visualized due to metallic artifacts on the CT images or when creating the CTA (n = 2). Ultimately, 41 patients (22 patients with conventional HRCT and 19 with U-HRCT) were enrolled in this study. Of the 22 patients scanned on the conventional HRCT scanner, 12 were male and 10 were female. The median age was 63 years old (range 39 to 87). The indications for cervical CTA with conventional HRCT were as follows: cancers in the oral cavity (n = 7); oropharyngeal cancer (n = 7); cancers in the nasal cavity or paranasal sinuses (n = 4); metastasis of a cervical lymph node(s) (n = 2); external ear cancer (n = 1); and lacrimal gland cancer (n = 1). Of the 19 patients scanned on the U-HRCT scanner, 18 were male and one was female. The median age was 61 years old (range 42 to 81). The indications for cervical CTA with U-HRCT were as follows: cancers in the oral cavity (n = 2); nasopharyngeal cancer (n = 2); oropharyngeal cancer (n = 6); cancers in the nasal cavity or paranasal sinuses (n = 6); metastasis of a cervical lymph node (n = 1); external ear cancer (n = 1); and angiosarcoma of the scalp (n = 1). 

#### 2.2.2. CT Scan Parameters and Method

For conventional HRCT, the scanning and reconstruction parameters were as follows: tube voltage = 120 kVp; tube current = automatic exposure control (AEC); collimation = 0.5 mm × 80 slice; rotation time = 0.5 s; beam pitch = 0.806 or 0.828; imaging field of view (FOV) = 230 mm; reconstruction kernel = FC04 (for mediastinum); iterative reconstruction = AIDR3D; matrix = 512 × 512; slice thickness = 0.5 mm.

For U-HRCT, the scanning and reconstruction parameters were as follows: tube voltage = 100 or 120 kVp; tube current = AEC (n = 17) or fixed current (260 mA, n = 1; 310 mA, n = 1); [SHR mode] collimation = 0.25 mm × 160 slice (n = 8) and [HR mode] 0.5 mm × 80 slice (n = 11); rotation time = 0.5 sec; beam pitch = 0.806 or 0.828; imaging field of view (FOV) = 230 mm; reconstruction kernel = FC04 (for mediastinum); iterative reconstruction = AIDR3D-e; matrix = 512 × 512; [SHR mode] slice thickness = 0.25 mm (n = 8) and [HR mode] slice thickness = 0.5 mm (n = 11).

Two types of contrast medium were administered, Iopamidol (Iopamiron 370: Bayer, Osaka, Japan) or Iohexol (Omnipaque 350: GE Healthcare Pharma, Tokyo, Japan). The CT scans were performed using the following protocol. First, 50 mL of contrast medium was injected, followed by 30 mL of saline at 4–5 mL/s. The trigger threshold was set at 150 Hounsfield units (HU) for the round region of interest (ROI) in the aortic arch. After the HU value of the ROI reached the threshold, the scanning and data acquisition were started by the radiology technologists. The CT scanning started from the frontal sinus and finished below the aortic arch.

Radiation exposure was assessed by the CTDIvol and DLP, which were provided by the scanner in a dose information report for each patient. 

#### 2.2.3. Three-Dimensional CTA Preparation

Identical to the phantom study, all images of 3D cervical CTA were created using a single research workstation (Ziostation2). Unnecessary tissues for the 3D-CTA such as bones, soft tissue, and veins were automatically removed by the workstation using a density-masking technique. Furthermore, since this study targeted the external carotid arteries and their branches only, the internal carotid arteries and their branches were semi-automatically removed on the workstation by manually enclosing the origin and peripheral trunks of the internal carotid arteries. These processes were performed by two radiologists with 4 and 19 years of experience in head and neck imaging. 

#### 2.2.4. Arterial Visibility Assessment

The 18 arteries visualized bilaterally by 3D-CTA were evaluated in consensus by three radiologists with 4, 8, and 19 years of experience, who were unaware of scanning method, using a 5-point scale (score 0 = invisible; score 1 = the origin of the artery was only visualized; score 2 = the whole trunk of the artery was visualized but the branch was not visualized; score 3 = one or two branches of the artery were visualized; score 4 = three or more branches, or secondary branch(es) were visualized). The 18 arteries consisted of the following 9 arteries bilaterally (9 arteries × 2 sides): superior thyroid, lingual, facial, occipital, ascending pharyngeal, posterior auricular, maxillary, superficial temporal, and transverse facial arteries.

### 2.3. Statistical Analysis

Regarding the phantom study, the comparison between conventional HRCT and two scan modes of U-HRCT was performed by one-way analysis of variance (ANOVA) and Bonferroni’s multiple comparison test for density measurement and by Friedman’s test and Scheffe’s test for visual assessment, respectively. Comparisons in the patient study between conventional HRCT and two scan modes of U-HRCT were made by one-way analysis of variance (ANOVA) and Bonferroni’s multiple comparison test for radiation exposure dose and by the Kruskal–Wallis test and Steel–Dwass test for visual assessments, respectively. A *p*-value of <0.05 was the threshold for significance. All statistical analyses were performed using BellCurve for Excel (version 2.15).

## 3. Results

### 3.1. Phantom Study

#### 3.1.1. Radiation Dose Assessment

The CT Dose Index volume (CTDIvol) was 7.0 mGy, 7.8 mGy, and 7.7 mGy for conventional HRCT, SHR mode, and HR mode, respectively. The Dose Length Product (DLP) value was 244.7 mGy⋅cm, 271.9 mGy⋅cm, and 269.8 mGy⋅cm for conventional HRCT, SHR mode, and HR mode, respectively.

#### 3.1.2. Arterial Density Assessment

Comparisons of the maximum density values of the mock arteries between conventional HRCT and two scan modes of U-HRCT are shown in Table 1 and Figure 1. For the proximal artery, the maximum density of the SHR mode was significantly higher than that of conventional HRCT (*p* < 0.01). For the peripheral artery, the maximum density of the SHR mode was significantly higher than that of conventional HRCT or the HR mode (*p* < 0.01).

#### 3.1.3. Arterial Visibility Assessment

Comparisons of the visual assessment of the mock arteries between conventional HRCT and two scan modes of U-HRCT are shown in Table 2 and Figure 2. The score of the SHR mode was significantly higher than that of conventional HRCT (*p* = 0.034).

### 3.2. Patient Study

#### 3.2.1. Radiation Dose Assessment

In this study, the mean CTDIvol value was 9.22 ± 2.38 mGy for conventional HRCT, 9.96 ± 3.70 mGy for the SHR mode, and 9.67 ± 3.04 mGy for the HR mode, respectively. The mean DLP value was 366.0 ± 120.5 mGy⋅cm for conventional HRCT, 384.6 ± 119.9 mGy⋅cm for the SHR mode, and 375.9 ± 111.9 mGy⋅cm for the HR mode, respectively. No significant differences in CTDIvol and DLP were found between each of the two groups.

#### 3.2.2. Arterial Visibility Assessment

The qualitative scores are summarized in Table 3. In 17 of 18 arteries, the mean score of the SHR mode was significantly higher than that of conventional HRCT (*p* < 0.05). Additionally, in all arteries, the mean score of the HR mode was significantly higher than that of conventional HRCT (*p* < 0.05). Particularly, the differences in the scores were greater for smaller arteries (i.e., the ascending pharyngeal artery and the transverse facial artery) (Figure 3 and Figure 4). No significant difference was obtained between the SHR and HR modes.

## 4. Discussion

In this study, the analysis of cervical CTA for human subjects demonstrated that all cervical arteries from the external carotid arteries were more clearly depicted by U-HRCT (SHR and HR modes) than by conventional HRCT. No significant differences in radiation exposure were found between U-HRCT (SHR and HR modes) and conventional HRCT. This was partly supported by the phantom study, which found that the maximum density values of the small peripheral arteries were significantly higher with the SHR mode of U-HRCT than with conventional HRCT and that the score of the SHR mode at the peripheral level in visual assessment of CTA-like images was significantly higher than that of conventional HRCT. These results strongly suggest that for cervical CTA, more peripheral, smaller arteries are more accurately delineated by the SHR mode of U-HRCT. Although this study is the first to demonstrate the advantage of U-HRCT for the improvement of cervical CTA, several studies regarding the comparison of small arteries between the U-HRCT and conventional HRCT scanners have already been reported [20,21,22,23,24,25,26,27,28,29]. For example, Yoshioka et al. reported that the artery of Adamkiewicz is better depicted on U-HRCT than on conventional HRCT [20]. Takagi et al. reported that coronary CTA by U-HRCT demonstrates a better correlation with invasive coronary angiography than with conventional HRCT [21]. Murayama et al. reported that the visualization of the lenticulostriate arteries is improved by U-HRCT [23]. Our results suggesting the improvement of cervical CTA by U-HRCT are concordant with those of these previous reports.

Our unexpected result was that the score of the HR mode of U-HRCT in the visualization of all cervical arteries was significantly higher than that of conventional HRCT in the patient study. The matrix and the slice thickness of both scanning methods are the same; however, the channel number of the HR mode was twice in comparison with conventional HRCT. The increase in sampling number can contribute to the improvement in image quality for the HR mode. Although the phantom study did not show significant superiority of the HR mode against conventional HRCT, the mean maximum density of mock arteries and the visualization score were higher for the HR mode. Another plausible reason may be that the HR mode was selected when a patient was heavyset, that is, the length of the scanning range in the z-axial direction elongated. Therefore, the actual diameters of target branches from external carotid arteries for patients scanned with the HR mode might be larger than those for patients with the SHR mode. However, such patients should have been included in the group of conventional HRCT as well. The difference in patients’ habitus between groups of conventional HRCT and the HR mode would be not large. 

We believe that the use of U-HRCT scanners can offer advantages for planning intra-arterial infusion chemotherapy for malignant head and neck tumors. In addition, U-HRCT can be scanned without an increase in radiation exposure. Improvement of the visualization for cervical CTA enables better preoperative simulations by obtaining a more detailed vascular anatomy of a targeted tumor and more effective procedures with the choice of appropriate devices such as catheters and guidewires, resulting in a reduction in the cost of the operation and a reduction in radiation exposure by decreasing the time for the procedure and operation. Moreover, it would be useful to detect fine and/or complicated feeding arteries of a tumor, particularly for patients with arterial anomalies or precedent surgery that causes altered arterial connections. In addition, dangerous extracranial–intracranial anastomosis may be present in peripheral subdivisions. Accurate preoperative understanding of peripheral arteries is important to improve the effectiveness and safety of treatment. Although the detection of feeding arteries by U-HRCT was not evaluated in this study due to high inter-patient variability, this is an item to be studied in the future.

This study has several limitations. The first major limitation is the small patient population. Although our results showed the visualization of cervical CTA with U-HRCT was clearer than with conventional CTA, studies with larger patient populations are needed to confirm our findings. Second, the patient groups for U-HRCT and conventional HRCT were different. Individual differences in arterial structures may have affected the results. However, it is not possible to compare the same population in clinical practice; thus, our approach in the present study was practical. Third, the phantom used in this study was a head–neck CT phantom, which was not designed for the investigation of cervical CTA. Since no neck phantom was available with the mock external carotid arteries, we believe that our approach is justified. Finally, although we applied the unified method for contrast administration and scan timing, two kinds of contrast with slightly different iodine concentrations were administered. This was due to a change in the type of contrast media routinely applied at our facility during the study period. In addition, the arterial density of each patient might be different due to diversity in circulation dynamics. 

## 5. Conclusions

U-HRCT provides higher image quality in terms of visualization of arteries than conventional HRCT in cervical CTA.

## Figures and Tables

**Figure 1 cancers-16-03866-f001:**
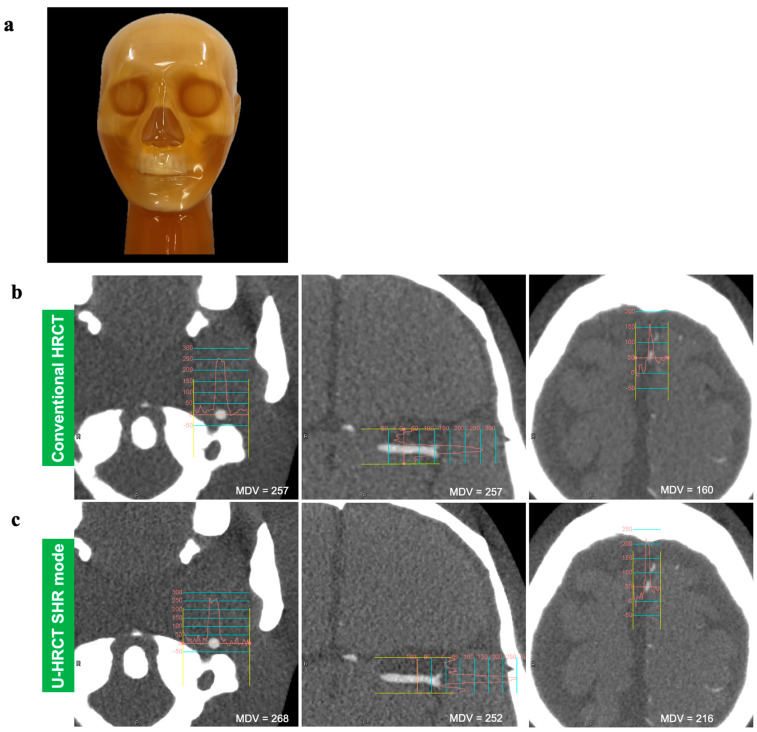
The actual appearance of a commercially available head-neck CT phantom (**a**). The density curves of the proximal, intermediate, and peripheral parts of the mock artery on axial images of conventional HRCT (**b**) and U-HRCT (SHR mode) (**c**). Note that the maximum density value (MDV) is similar at the intermediate part, while the MDVs of the proximal and peripheral parts of U-HRCT (SHR mode) are higher (268 HU and 216 HU) than those of conventional HRCT (216 HU and160 HU).

**Figure 2 cancers-16-03866-f002:**
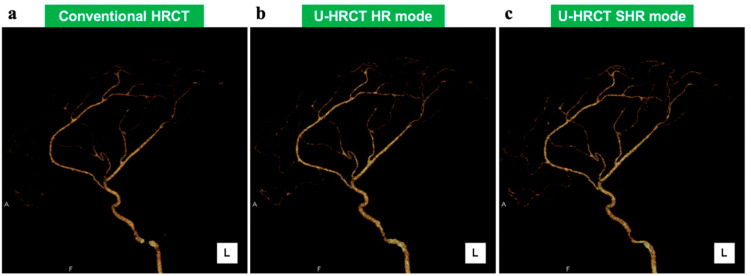
The 3D-CTA of conventional HRCT (**a**) and two scan modes of U-HRCT ((**b**) HR mode, (**c**) SHR mode). The visualization of all peripheral branches is obviously clearer by SHR mode (**c**) than by conventional HRCT (**a**). These images were displayed with window width of 116 and window length of 79.

**Figure 3 cancers-16-03866-f003:**
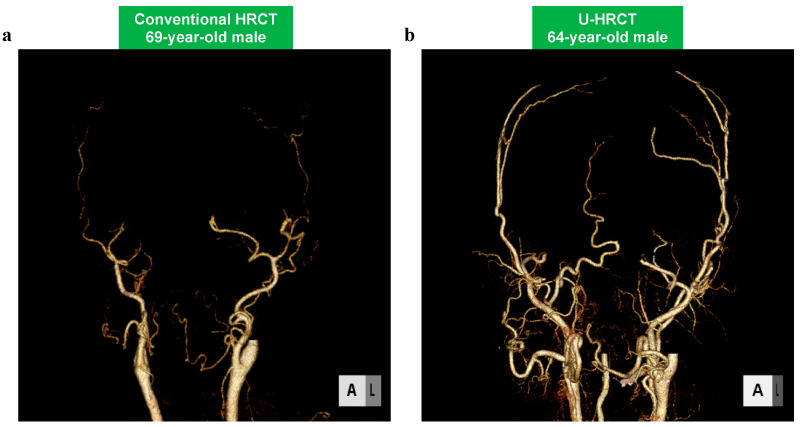
The 3D-CTA of the bilateral ECA (external carotid artery) and branches on conventional HRCT (**a**) and U-HRCT (SHR mode) (**b**). More arteries are clearly depicted by U-HRCT (SHR mode) (**b**).

**Figure 4 cancers-16-03866-f004:**
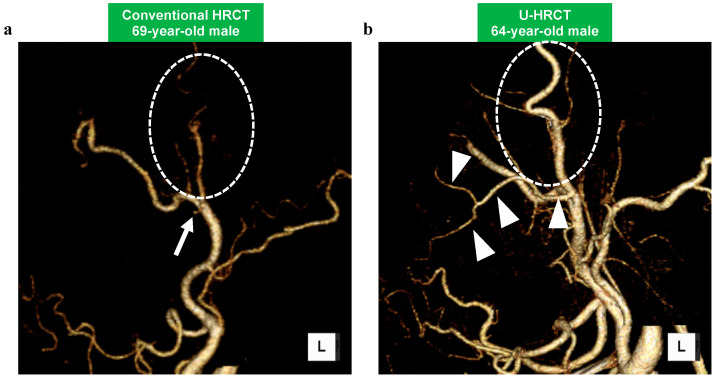
The 3D-CTA of the left transverse facial artery. While the origin of the artery is only visualized on conventional HRCT ((**a**) arrow), the peripheral parts and secondary branches are visualized on U-HRCT (SHR mode) ((**b**) arrowheads). Note that the visualization of the superficial temporal artery (ovoid circles) is much clearer by U-HRCT (SHR mode) (**b**) than by conventional HRCT (**a**). The visual scores of these are as follows: (**a**) left transverse facial artery = 1, left superficial temporal artery = 3; (**b**) left transverse facial artery = 4, left superficial temporal artery = 4.

**Table 1 cancers-16-03866-t001:** Comparison for maximum vessel density of the mock arteries in the phantom.

	Conventional HRCT (HU)	SHR Mode of U-HRCT (HU)	HR Mode of U-HRCT (HU)	*p* Value
**Artery**				
Proximal	266.1 ± 4.2	283.7 ± 12.1 *	277.0 ± 18.0	<0.01
Intermediate	254.5 ± 13.5	259.4 ± 26.6	259.7 ± 24.9	0.77
Peripheral	169.8 ± 25.8	220.7 ± 23.8 **	180.6 ± 16.1	<0.01

Abbreviations: HRCT = high-resolution computed tomography; U-HRCT = ultra-HRCT; HU = Hounsfield unit. Note: * The maximum density of SHR mode was significantly higher than that of conventional HRCT (*p* < 0.01). ** The maximum density of SHR mode was significantly higher than those of conventional HRCT and HR mode (*p* < 0.01).

**Table 2 cancers-16-03866-t002:** Comparison for visual assessment of mock arteries in the phantom.

Score	4	3	2	1
Conventional HRCT	5	2	1	2
SHR mode of U-HRCT	10	0	0	0
HR mode of U-HRCT	7	3	0	0

Abbreviations: HRCT = high-resolution computed tomography; U-HRCT = ultra-HRCT. Note: A total of 10 branches were evaluated. The score of SHR mode was significantly higher than that of conventional HRCT (*p* = 0.034).

**Table 3 cancers-16-03866-t003:** Comparison of the visual scores (visibility) for cervical arteries demonstrated by 3-dimensional CT angiography in the patient study.

**Right**	**Conventional** **HRCT**	**SHR Mode of** **U-HRCT**	**HR Mode of** **U-HRCT**	** *p* ** **Value**
**Artery**				
Superior thyroid	3.00	3.88 *	3.73 **	<0.01
Lingual	2.59	3.88 *	3.82 **	<0.001
Facial	2.55	3.88 *	4.00 **	<0.001
Occipital	2.77	3.75 *	3.91 **	<0.001
Ascending pharyngeal	1.59	3.13 *	2.82 **	<0.001
Posterior auricular	1.91	3.38 *	2.91 **	<0.001
Maxillary	3.09	4.00 *	3.82 **	<0.001
Superficial temporal	2.68	3.50 *	3.77 **	<0.001
Transverse facial	1.64	3.38 *	2.82 **	<0.01
**Left**	**Conventional** **HRCT**	**SHR Mode of** **U-HRCT**	**HR Mode of** **U-HRCT**	** *p* ** **Value**
**Artery**				
Superior thyroid	2.50	3.38	3.55 **	<0.01
Lingual	2.73	3.75 *	3.82 **	<0.001
Facial	2.50	4.00 *	4.00 **	<0.001
Occipital	2.82	4.00 *	3.82 **	<0.001
Ascending pharyngeal	1.59	3.25 *	2.82 **	<0.001
Posterior auricular	1.82	3.13 *	2.73 **	<0.01
Maxillary	3.18	4.00 *	3.64 **	<0.001
Superficial temporal	2.68	3.63 *	3.64 **	<0.001
Transverse facial	1.55	2.88 *	2.91 **	<0.01

Abbreviations: HRCT = high-resolution computed tomography; U-HRCT = ultra-HRCT; HU = Hounsfield unit. Note: * The mean score of SHR mode was significantly higher than that of conventional HRCT (*p* < 0.05). ** The mean score of HR mode was significantly higher than that of conventional HRCT (*p* < 0.05). No significant difference was obtained between SHR and HR modes.

## Data Availability

The data presented in this study are available on request from the corresponding author. The data are not publicly available due to privacy restrictions.

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
