# Peer review of "Cervical CT Angiography: The Advantage of Ultra-High-Resolution CT Versus Conventional HRCT"

_cancers, 2024, doi:10.3390/cancers16223866_

Round 1

Reviewer 1 Report

Comments and Suggestions for Authors

The paper deals with the use of a new CT with a higher resolution which may be more suitable for studying the arteries of the head and neck. The article is well written and the results are presented correctly.

Minor remarks

There are various places where there are white spaces between the slice thickness and, I assume, the number of slices (lines 85,912,144,149). In my opinion the space should be completed with an X and writing “slice”.

Phantom study.

It would be better to insert a photo of the phantom, and explain better how it is made, because it is not clear.

Author Response

Comments 1:

Minor remarks

There are various places where there are white spaces between the slice thickness and, I assume, the number of slices (lines 85,912,144,149). In my opinion the space should be completed with an X and writing “slice”.

Response 1:

Thank you very much for invaluable comments. I changed some spaces to “x” between the slice thickness and the number of slices and added “slice” after the thickness (line 86, 92, 93, 149, 154 and 155).

Comments 2:

Phantom study.

It would be better to insert a photo of the phantom, and explain better how it is made, because it is not clear.

Response 2:

I explained the material of the head phantom on line 78-79 and inserted a photo of the phantom in Figure 1 (a) on page 6.

Reviewer 2 Report

Comments and Suggestions for Authors

General comments

This study assessed the advantage of ultra-high-resolution computed tomography (U-HRCT) over conventional high-resolution computed tomography (HRCT) for cervical CTA from the viewpoints of both phantom and patient studies. The authors concluded that U-HRCT provides higher image quality in the visualization of arteries than conventional HRCT in cervical CTA. This study was conducted in a well-structured style, and the description seems sound, providing evidence of the value of U-HRCT in CTA of the cervical region. If the authors satisfactorily respond to specific comments and instructions from the reviewers, the manuscript could be accepted in this journal.

Specific comments

Simple Summary and Abstract

1.       In the Simple summary, the last sentence stating the conclusion of this study does not describe the actual conclusion of the authors. It is recommended that the sentence be changed to a similar sentence to that of Abstract.

2.       In “Background/Objectives” of the Abstract, please briefly include the background of this study referring to intra-arterial infusion chemotherapy.

Introduction

3.       In the first sentence of the second paragraph, “has been clinically applied” is to be “has been clinically applied”.

Materials and Methods

4.       In the first part, it is unclear when in this study the participants were provided with an opt-out opportunity. Please clarify.

5.       In both the phantom study and the patient study, it is likely that, judging from Figures 2, 3, and 4, the visual evaluation was performed on volume-rendered CTA images, not on MIP images. This should be clearly stated referring to the reason(s) why the authors selected volume-rendered images.

6.       In the patient study, it is unfortunate, as the authors admit in the Discussion as one of the limitations of this study, that the two kinds of iodinated contrast agents were utilized. It is strongly recommended to describe the reason.

Results

7.       Table 3 shows the visual scores for cervical arteries. The figures shown in this table are the mean scores. What does “(HU)” in this table represent?

Discussion

8.       I agree with the statement that the use of a U-HRCT scanner can offer advantages for planning intra-arterial infusion chemotherapy for malignant head and neck tumors. However, is it necessary to visualize the distal part of major branches of the external carotid artery that are to be selectively catheterized in infusion chemotherapy? This issue should be additionally discussed briefly.

9.       In addition, it is somewhat unfortunate that the authors just focused on the delineation of branches of the external carotid artery without evaluating the delineation of lesions per se. This issue also needs to be discussed.

Author Response

Simple Summary and Abstract

Comments 1:

In the Simple summary, the last sentence stating the conclusion of this study does not describe the actual conclusion of the authors. It is recommended that the sentence be changed to a similar sentence to that of Abstract.

Response 1:

Thank you very much for very invaluable comments. I deleted the last sentence in the Simple summary and changed to same sentence to that of Abstract (line 22-23).

Comments 2: 

In “Background/Objectives” of the Abstract, please briefly include the background of this study referring to intra-arterial infusion chemotherapy.

Response 2:

We added the following sentence to the first of Background/Objectives of Abstract (line 24-25). “Pre-treatment depiction of the cervical arteries is important for better intra-arterial infusion therapy of malignant head and neck tumors.”

Introduction

Comments 3:

 In the first sentence of the second paragraph, “has been clinically applied” is to be “has been clinically applied”.

Response 3:

Thank you for pointing out of our grammar error. We corrected the sentence (line 57).

Materials and Methods

Comments 4:

In the first part, it is unclear when in this study the participants were provided with an opt-out opportunity. Please clarify.

Response 4:

We clearly stated that the opt-out opportunities were provided to the participants on the website of our department (line 72-74).

Comments 5:

In both the phantom study and the patient study, it is likely that, judging from Figures 2, 3, and 4, the visual evaluation was performed on volume-rendered CTA images, not on MIP images. This should be clearly stated referring to the reason(s) why the authors selected volume-rendered images.

Response 5:

We selected volume-rendered 3D CTA rather than MIP because we believe that 3D CTA might be easier to understand in terms of three-dimensional anatomical structure. We added the following sentence on line 111-113. “We selected 3D CTA-like image, which might be easier to understand in terms of location and anatomical structure than maximum intensity projection image.”

Comments 6:

In the patient study, it is unfortunate, as the authors admit in the Discussion as one of the limitations of this study, that the two kinds of iodinated contrast agents were utilized. It is strongly recommended to describe the reason.

Response 6:

This was due to a change in the type of contrast media routinely applied at our facility during the study period. We added this sentence on line 338-339.

Results

Comments 7:

Table 3 shows the visual scores for cervical arteries. The figures shown in this table are the mean scores. What does “(HU)” in this table represent?

Response 7:

In Table 3, “(HU)” was inserted incorrectly and we deleted all of that.

Discussion

Comments 8:

I agree with the statement that the use of a U-HRCT scanner can offer advantages for planning intra-arterial infusion chemotherapy for malignant head and neck tumors. However, is it necessary to visualize the distal part of major branches of the external carotid artery that are to be selectively catheterized in infusion chemotherapy? This issue should be additionally discussed briefly.

Response 8:

We believe that it would be useful to detect fine and/or complicated feeding arteries of a tumor, particularly for patients with arterial anomalies or precedent surgery that causes altered arterial connections. In addition, dangerous extracranial-intracranial anastomosis may be present in peripheral subdivisions. That is why accurate preoperative understanding of peripheral arteries is important to improve the effectiveness and safety of treatment. We added similar sentences on line 320-325.

Comments 9:

In addition, it is somewhat unfortunate that the authors just focused on the delineation of branches of the external carotid artery without evaluating the delineation of lesions per se. This issue also needs to be discussed.

Response 9:

Thank you very much for very important suggestions. The detection of feeding arteries by U-HRCT could not evaluated due to high inter-patient variability in this study. We considered this should be studied in the future. We added sentences about this on line 325-326.